# Global and Brazilian Scenario of Guidelines and Legislation on Welfare in Pig Farming

**DOI:** 10.3390/ani12192615

**Published:** 2022-09-29

**Authors:** Isabella Cristina de Castro Lippi, Fabiana Ribeiro Caldara, Ibiara Correia de Lima Almeida Paz, Agnês Markiy Odakura

**Affiliations:** 1School of Veterinary Medicine and Animal Science, São Paulo State University, Street Prof. Dr. Walter Maurício Corrêa w/n, Botucatu 18618-687, SP, Brazil; 2Faculty of Agricultural Science, Federal University of Grande Dourados, Itahum Highway, km 12, Dourados 79804-970, MS, Brazil

**Keywords:** animal welfare, ethics, laws, sentience

## Abstract

**Simple Summary:**

As with all production animals, pigs need environments suitable for their species, which results in less stress and suffering and more productivity. Therefore, there are quality standards and good management and production practices that must be followed to ensure their well-being. In this sense, over the last few years, many advances have been made, banning or restricting practices capable of causing suffering to animals in most producing countries. This review seeks to explore current global and Brazilian regulations on good production practices in swine farming.

**Abstract:**

The evolution of scientific knowledge regarding animal sentience, together with the growing concerns of consumers regarding current production models, has brought with it the responsibility of reviewing many practices carried out in industrial swine farming, with the purpose of improving the life quality of animals throughout the entire production cycle. In this sense, many initiatives have been taken by European Union, OIE and other countries to abolish questionable practices from an animal welfare point of view, being signed through legislation or normative instructions, which guide governments and companies on the best practices to be adopted. Among the main changes that have taken place in swine farming are the ban or reduction in the use of cages for sows, restrictions on the age at weaning, ban on painful procedures such as surgical castration, tail and teeth clipping, as routine procedures or without the use of anesthesia/analgesia. In addition, these acts also prescribe practices that must be adopted in order to respect the natural behavior of animals, such as the use of environmental enrichment. This review aims to address the main advances made over the last few years in the protection of swine, as well as Brazilian initiatives in this regard.

## 1. Introduction

The intensification in pig production has enabled better monitoring of animals, leading to lower energy losses, space optimization, increased productivity, and, consequently, lower production cost. However, modern production systems bring about severe alterations in the inherent behaviors of pigs, in addition to subjecting them to questionable procedures from the viewpoint of animal welfare [1]. 

Throughout the productive cycle, animals are exposed and submitted to many potentially stressful situations such as early weaning, regrouping, high stocking rates, chronic hunger (pregnant sows), low air quality, and invasive procedures such as castration, tail docking, and tooth clipping [2]. Animals reared under artificial and restrictive conditions are more prone to stress and often exhibit undesirable behaviors, which entails debates on the management adopted and confinement conditions in production systems, which are directly associated with poor welfare [3]. 

For those reasons, the debate on animal welfare has become prominent worldwide, driving a growing number of aware consumers concerned with the physical and mental integrity of production animals. Therefore, animal welfare represents, along with sustainability, one of the main challenges for the sector. 

Society pressure demands the development of researches and methodologies that enable an adequate level of welfare allied with high productivity and profitability of the system. From this viewpoint, several countries seek to adapt to the new market trends and requirements by creating legislation, decrees, and administrative acts on the welfare of production animals. 

The European Community has pioneered legislation on the matter and enforced minimum criteria of housing and management [4]. According to Ingenbleek et al. [5], ethics issues regarding the current production methods must be revised in countries that produce and, in particular, export pork. Legislations on animal welfare adopted in some countries, such as those in the European Union, are relevant in the global productive scenario as they foster changes in the way exporting countries carry out production [6].

For proper analysis of animal welfare, all factors that have the potential to directly impact animal life from birth to slaughter must be considered, such as the emotions it experiences, organism working, and interaction with the environment the animal is inserted in [3]. From a practical standpoint, positive welfare may be achieved by keeping animals in an adequate environment for their rearing that allows them to express as many natural behaviors and aspects as possible, in addition to care with nutrition and health and adequate treatment by handlers [7]. 

Some actions that can be taken to favor welfare in pig production systems stand out: eliminating factors that directly impact welfare, such as surgical castration without anesthesia or analgesia and cages for pregnant sows, and adopting practices that help mitigate stress, such as environmental enrichment. An animal that is able to express a broad range of positive behaviors will be more likely to deal with stressful events around it [8]. 

European countries have been discussing animal rights for about 200 years and their pioneering began in 1822 with the implementation of Martin’s Act in Great Britain, whose basis was the prevention of cruelty and improper treatment of cattle in production systems. The first community legislation of the EU was adopted in 1974, determining the obligation of pre-slaughter stunning in mammals. Subsequently, other norms were established involving all species of productive interest, covering production, transport, slaughter and experimentation. Specifically for pigs in intensive farming systems, the Directive 2008/120/EC is the legislation used in the EU, which places European pig farming at the forefront of animal protection. It should be noted that each country belonging to the bloc can establish its own rules, as long as they are stricter in relation to this Directive. 

In Brazil, despite the scarce specific legislation on animal welfare, administrative acts concerning this issue have been published by the Ministry of Agriculture, Livestock, and Food Supply (MAPA) and have led to some advances. Normative Instruction 12/2017 establishes guidelines for the accreditation of entities to carry out training in pre-slaughter and slaughter of animals [9], whereas Ordinances 524/2011 and 905/2017, respectively, created the permanent technical commission on animal welfare (CTBEA) [10]. Nonetheless, such publications approach production species as a whole, which makes them superficial, i.e., they still lack specific and clear information [6]. Recently, in order to guide producers on the best alternatives to promote an increasingly sustainable and competitive pig industry, Normative Instruction 113 (16 December 2020) [11] was established by MAPA, which came into force in February 2021, regulating good management and animal welfare practices in commercial pig farms, and discusses numerous aspects that will be addressed throughout this review. 

In face of that, this review aims to present an outlook of the international and Brazilian scenarios concerning administrative acts, guidelines, decrees, and legislations on welfare in pig farming while discussing the most relevant aspects and practices in that context. The main questions and questionable practices in pig farming from the point of view of animal welfare will be addressed here, except those related to pre-slaughter and slaughter management. It is worth mentioning that, in many cases, despite the lack of specific legislation, many countries are committing themselves to adopt or exclude certain practices due to the pressure exerted by the large companies in the food sector in the world.

## 2. Gestation and Farrowing Cages

The gestation-lactation cycle of cage use results in precarious and stressful life conditions for the sows. In this type of housing, with diminute spaces, the sows are prevented from performing any innate behaviors and even turning around. Previously, at the end of gestation, the sows were transferred to farrowing crates and, soon after weaning, they were taken back to the gestation cages to be inseminated, where they spent the entire gestational period [12]. And so, they spent their entire productive lives confined in small individual cages. For this reason, this practice began to be refuted and banned in many countries and alternative systems were proposed, such as housing sows in groups during gestation, which provides them with greater freedom of movement and, therefore, better welfare [13].

Group gestation consists in management in which the sows are housed in pens and remain free to roam along with other females, which may be in the same gestational phase (static group gestation) or in different gestation phases (dynamic group gestation), which allows them to manifest natural behaviors and improves welfare. In this model, sows remain in the collective housing throughout the gestational period or for most of it, until they near delivery. Subsequently, females will usually be housed in farrowing crates to prevent piglet crushing [14].

According to directive 2008/120/EC, pregnant sows can remain in individual cages for a maximum period of 28 days, after which they must be transferred to collective pens. More recently, on 30 June 2021, the European Union Commission approved the project of a popular initiative of community citizens (European Citizens’ Initiative—ECI) entitled “End of the Cage Age”. From then on, the commission committed to establishing a transition plan, to be published by the end of 2023, promoting the gradual reduction of industrial animal husbandry in cages until the total ban from 2027 [15]. 

When it comes to advances in the implementation of measures towards pig welfare, Sweden exhibits the highest standards. With only 1.1% of the herd in the European Community, the country has abolished the use of gestation cages and mandated farmers adequate their facilities to house sows and gilts in group gestation warehouses. Those females must be collectively housed after the fourth day of mating and remain free to roam up until a week prior to their farrowing. Sweden has also prohibited the use of slatted floor for pigs of any phase and age. Following that example, Holland mandates females be free to roam four days after mating [16].

In Denmark, pregnant sows must not be kept in pens narrower than 3 m and must be housed in groups from piglet weaning to seven days prior to the next farrowing. Such legislation went into effect on 1 January 2015 for new constructions and must be complied with by all farms until 1 January 2035 both for production for the domestic market and for export to the United Kingdom. The United Kingdom, in turn, banned housing in individual gestation cages in 1999 and adopted similar requirements as Denmark [17].

Other countries in the European Union, such as Spain (Royal Decree 1135/2002) and Germany, follow the established conditions in the Directive 2008/120/EC, and allow sows to be kept in individual crates until the fourth week after insemination to ensure better gestation maintenance. After that period, the animals must remain in group gestation until a week prior to the due date [18]. 

Also, according to the guideline, sows kept in groups must have a total unobstructed area of at least 1.64 m^2^ per gilt and at least 2.25 m^2^ per sow. For groups of less than 6 individuals, the unobstructed floor must have a 10% increase in its area. In case of groups larger than 40 individuals, the unobstructed floor area can be reduced by 10%. The sides of pens for pregnant sows must be more than 2.8 m long. The sides of pens for pregnant sows housed in groups of less than six individuals must be more than 2.4 m long. Part of the above-mentioned unobstructed floor area must be continuous. Specifically, 0.95 m^2^ of continuous floor must be available for each gilt after service and 1.3 m^2^ for each sow. In both cases, a maximum of 15% of the continuous floor must be reserved for drainage spaces [19].

In other continents, several countries have also adapted to incorporate changes towards the welfare of their breeding herds. Canada prohibited in 2014 the use of continuous confinement crates in pig farming. The measure follows the Code of Practice for the Care and Handling of Pigs and farms must comply until 2024. However, the National Farm Animal Care Council (NFACC) revised the code of practice in 2019 and extended the deadline for adapting to the loose system to 2029 [20].

Some states of the United States, such as California, Arizona, Colorado, Florida, Maine, Oregon, Ohio, Michigan, and Rhode Island have banned the use of individual gestation crates, whereas the other states allow such housing throughout gestation [21]. On 9 March 2022, the US Congress was presented with the bill “H.R. 7004: PIGS Act of 2022” to amend the “Animal Welfare Act (7 U.S.C. 2131)” to prohibit the confinement of pregnant pigs, and for other purposes, throughout the national territory. According with this bill it shall be unlawful for a person to cause any breeding pig to be confined in—“(A) such a manner that prevents the pig from lying down, standing up, or turning around:“(i) in a complete circle without any impediment, including a tether; and “(ii) without touching the side of an enclosure or another animal; and “(B) a space with less than 24 square feet of usable floorspace per pig. At this moment (August 2022) bill is in the in the stage of the legislative process.

Australia is a large country that has several peculiarities, with environmental, economic, and social conditions that vary between jurisdictions. Animal welfare legislative frameworks are a product of jurisdiction locality and associated geography. In Australia, each state and territory regulate the handling of farm animals in accordance with the livestock statutes, which contain animal welfare provisions [22]. Australia does not yet have legislation banning sow gestational crates. Despite the lack of laws prohibiting the practice, approximately 80 percent of the Australian sow herd is installed “sow stall free” and products bearing the “sow stall” label are widely marketed in supermarkets. In late 2016, the Australian Pork Limited (APL) requested a law banning the use of barns by the country’s swine producers [23].

In New Zealand, the practice of confining sows was discontinued in 2015 [24], and the committe believes that the use of farrowing crates should also be phased out but recognises this can only happen when alternative management systems and technologies are in place.

Early in the debate on animal welfare, large companies feared such changes in favor of better rearing conditions would pose a risk to business in face of increased production costs and the publicizing of sensationalist images by the media and non-governmental organizations [25]. In this context, the study Benchmark, carried out in 2016 with 110 companies in the food sector, reported that 79% of them had committed to avoiding confinement systems with restricted space in at least one of the markets in which they were present [25].

In Brazil, the IN 113/2020 [11] determines that keeping females in individual gestation cages is tolerated and limited to 35 (thirty-five) days after insemination. In addition, it determines that the cages used for reproductive management, insemination and weaning-estrus interval must be properly sized to allow the females to rest without simultaneously touching both sides of the cage and to stand up without touching the upper and side bars of the cage. According to this normative instruction, farms that use gestation cages and cages for boars will have until 1 January 2045, to adapt their facilities for collective gestation and pens for males. For new projects, filed with an environmental agency, with the prior license in progress, the term for the adjustments will be 10 years. At the same time, renowned companies such as BRF, JBS, Frimesa, and Aurora, have committed to abolishing the use of individual crates and adequating all their owned farms to the group gestation model until 2026 [6].

In the UK and most parts of the European Union, pregnant sows can be kept in farrowing crates from the 7th day before their due date. However, they must have access to material that allows them to express natural nesting behavior (straw, hay, wood, sawdust, mushroom compost, peat or a mixture of these). The only exception is when the use of these materials could cause damage to the waste system. Those crates must have an exclusive area for piglets with proper temperature and a heating source.

The goal of the Danish industry was for 10% of sows to be released in farrowing pens until 2020. Starting in 2021, all new facilities must be designed with free-roam farrowing systems. Switzerland prohibits the use of farrowing crates, but allows their temporary use during delivery in some exceptional cases. In countries such as Canada, and the USA, the use of farrowing crates is legally allowed, with animals housed in them during delivery and lactation.

In Brazil, the use of farrowing crates is tolerated, but they must comply with the same requirements in relation to the dimensions reported for pregnant sows, and equipped with a heating source for newborn piglets. In order to get used to the environment, the sows must be transferred to the farrowing crates at least two days before the expected date of delivery, being considered the resource of enrichment appropriate to the nesting behavior prior to delivery [11].

## 3. Weaning

European Union countries such as Denmark, Holland, and Germany, according to Directive 2008/120/EC, allow piglet weaning with 28 days or 21 days for batch farrow production. One exception is Sweden, which prohibits weaning piglets younger than four weeks (27 days) [26]. In Brazil, the rural property must have written technical guidance for the weaning period in order to minimize the stress on piglets and sows, and new projects or expansion of farms must be designed for weaning the lot average age of 24 days or more. The farms that currently wean piglets with an average age of 21 days have until 1 January 2045 to adapt their facilities [11].

## 4. Procedures in Piglets

Piglets in modern rearing systems are subjected to a number of procedures during the first few days or weeks of life, including teeth clipping, tail docking, ear carving or another method of identification (tattooing), and perhaps most controversially, surgical castration without anesthesia and analgesia. All of these procedures involves some degree of tissue damage resulting in the piglet experiencing pain. As justifications for the routine execution, these management practices are based on hypotheses of economic interest and related to the improvement of well-being in some aspects, among them the decrease in lesions caused on the udders of lactating sows (due to the teeth clipping), the prevention of tail biting (minimized by tail docking) and reducing the aggressiveness of the castrated males after the nursery stage. In addition, they can be considered the advantages that these managements result, such as, for example, the castration of males, which leads to less depreciation of the carcass or, in the case of the tail docking, to reducing damage to parts of animal carcasses that have suffered as a result of tail biting [27].

### 4.1. Castration

The main purpose of male castration is to prevent the occurrence of unpleasant odor and flavor in the meat from the presence of androsterone and skatole [28]. Traditionally, the surgical procedure is one of the main methods used for pigs castration. However, when surgical castration is performed without anesthesia and analgesia is capable of inducing acute and chronic pain, in addition to behavioral changes, and promoting an acute activation of the sympathetic nervous system and the hypothalamic-pituitary-adrenal axis [29].

In the European Union, the surgical castration is allowed, but it must be carried out by a veterinarian and, in case it is performed after the 7th day of life, the professional must use anesthesia followed by extended analgesia [16].

A volunteer agreement dictates that surgical castration will be phased out starting in 2018. In the United Kingdom, the standards and requirements of Red Tractor, a quality seal licensed by the non-profit Assured Food Standards, does not allow castration and is an initiative to identify and value quality products in the country. For that reason, only 2% of male pigs are castrated, a significantly lower number compared to Sweden (94%), Holland (20%), Germany (80%), and Spain (20%) [30].

In Denmark, new legislation went into effect on 1 January 2018 allowing producers themselves, after properly trained, to apply local anesthesia for castration. According to the president of the Danish Agriculture & Food Council, Denmark once again went beyond the European Union legislation by setting the goal of ensuring that all male piglets would be given local anesthesia for the procedure until the end of 2018.

Australian regulations recommend that when surgical castration is necessary, it must be performed after the 2nd day and before the 7th day of life. When performed between the 8th and the 21st day, effective means of containment are required. After the 21st of life, only veterinarians can perform it, but with the use of anesthesia [31]. According to New Zealand legislation, it is desirable that surgical castration is not performed, but if indicated, it must be operated by Veterinarians, regardless of the age of the pig [24].

In Canada, surgical castration after the 10th day of life should be performed with anesthesia and analgesia. As of July 2016, the use of painkillers in castration at all ages became mandatory [20]. In the USA, there is no specific legislation on whether or not to use anesthetics or analgesics while performing the procedure.

In Brazil, through the Decree 9013 of 29 March 2017, prohibits the slaughter of intact males [32]. In this country, surgical castration is an accepted method, but starting February 2021, it can only be performed when recommended by a veterinarian and performed by a trained operator; equipment used with proper maintenance and sanitized; procedures have been adopted to minimize any pain, anguish and subsequent complications for the animal, according to the regulation of the Federal Council of Veterinary Medicine. However, the farms will have until 1 January 2030 to use analgesia and anesthesia, in any surgical castration, regardless of the animal’s age. Surgery to reduce scrotal hernia, vasectomy or other non-routine procedure can only be performed with no pain, using anesthesia and prolonged analgesia [11].

### 4.2. Tail Docking

Animals that suffer tail bites show physical damage and signs of fear. In addition to being more likely to have pleuritis (inflammation) and lung abscesses, their carcasses are also more likely to need trimmings. [33]. Tail docking helps control tail biting behavior, reducing the incidence and severity of injuries, but it doesn’t treat the causes of the problem. Regardless of the method used to tail docking, the procedure is stressful and results in acute pain [34].

Directive 2008/120/EC of 18 December 2008, establishes that this procedure should not be performed routinely. It is accepted only when it is possible to demonstrate that, despite taking all preventive measures, tail biting continues to be a problem for farm animals. Tail clipping performed after the seventh day of life can only be performed by a veterinarian under anesthesia and prolonged analgesia. The expression of natural behaviors, such as foraging, is one of the key factors in avoiding tail biting. For this reason, EU legislation determines that all pigs must have permanent access to a sufficient amount of foraging material, ensuring adequate investigation and handling activities. In Denmark, Sweden, Finland and Lithuania, there is specific legislation further limiting tail docking. In Denmark, when strictly required, docking must be performed between two and four days after birth [16,17,18,19,20,21,22,23,24,25,26,27,28,29,30]. and should be docked as little as possible and it is not allowed to dock more than ½ of the tail. In the United Kingdom and Holland, tail docking can be performed in the first 72 h of life, but, it must not be a routine procedure, while Sweden prohibits it since 1988 [35]. In Finland, cutting off an animal’s tail is prohibited as it is considered an act that causes unnecessary pain to the animal (law 2002:0910). Finally, in Lithuania, tail docking is completely prohibited [36].

In Switzerland and Norway, tail docking is strictly regulated. In the current Swiss regulations (Animal Protection Ordinance, Switzerland, 2001), tail docking of piglets can only be performed under anesthesia. Norwegian regulations state that tail docking can only be performed for medical reasons and can only be done by veterinarians with prolonged anesthesia and analgesia. (Regulation for Housing of Swine from 2003, § 10) [36].

Australia also recommends avoiding the tail docking, but when necessary, it must be done before the 7th day of life [20]. New Zealand limits this management until the 3rd day of life, allowing a maximum cut of half of the tail. If the cut has to be done after the 7th day of life, only Veterinarians can perform it [24].

In Canada, the docking should also be made only if it is proven necessary, and it should be performed up to 72 h of life, sectioning a minimum portion that preserves the coverage of the anus by the portion of the remaining tail. The tail cut after the 7th day must be performed with effective pain control. However, the new guidance states that as of July 2016, the use of analgesics in the tail cutting protocol will be mandatory for pigs of all ages [20].

The USA allow the procedure, however, they must follow guidelines on how management should be carried out [20]. In Brazil, tail docking should be avoided. However, the procedure will be allowed if the environmental adjustments provided for in the Normative Instruction are adopted, but the tail biting problem persists. For these cases, only the final third of the tail will be cut, up to the third day of life of the piglet. After three days of age, they will only be performed with the use of anesthesia and analgesics for pain control [11].

### 4.3. Tooth Clipping

Piglets are born with canine teeth sticking out, these teeth are used to compete with littermates for access to the ceiling. In the first week of life these teeth can be blunted or shortened by grinding the tip, clipping the tip, or clipping the tooth at the gum line. Regardless of the technique used in this management, the occurrence of injuries varies from the opening and exposure of the pulp cavity, to hemorrhages, infiltrations, abscesses, formation of osteodentin, to the tooth fracture. Most of these damages are more common and earlier when using pliers for this handling [37].

Tooth clipping in the European Union is allowed, but not as a routine procedure. Instead, it can only be performed after lesions have been verified in the teats or ears of sows and in the tail of other pigs [19]. This country considers partial tooth grinding as probable causes of immediate and extended pain in pigs. Therefore, measures must be enforced that ensure better practices. When tooth clipping is required, Denmark, the UK, and Holland establish it must be performed until the 3rd day of life, whereas Germany allows it until the 7th day.

Australia only allows the removal of the fourth part of the theeth, and the procedure must be performed in the first three days of life [20]. New Zealand, on the other hand, admits clipping teeth up to the 4th day of life, and recommends grinding instead of clipping [24]. Canada recommends that only piglets that show aggressive behavior should have their teeth cut, limiting removal to half the thoot [20].

In Brazil, the practice was legally allowed and performed around the second day of life of piglets. However, according to the new Normative Instruction, the tooth clipping is prohibited, and only grinding of the final third is allowed, when there is a serious injury to the mammary system of the matrix or the face of the litter piglets [11].

## 5. Feed Management

Countries of the European Union have determined, through [19], that all pregnant sows and gilts must be provided enough amounts of roughage with high fiber content, in addition to high-energy feed, so as to decrease hunger and fulfill their mastication needs.

Only Germany has a specific requirement towards decreasing the feeling of hunger of animals: The fiber content in the dry matter must be at least 8%, or a close value, to ensure the female has a daily fiber intake of at least 200 g.

## 6. Environmental Conditions and Environmental Enrichment

European Directive 2008/120/EC mentions the use of straw bedding in rearing systems to provide animal comfort. Over 90% of sows in Sweden are reared with straw bedding and only 1% of the pigs in that country are reared in free-range, organic systems. In Denmark, all pigs must have permanent access to sufficient amounts of straw or other enrichment materials. Such enrichment must be based on natural materials and be provided as bedding while the use of chains as the single form of enrichment in pens is not allowed. Holland and Germany require all animals have permanent access to materials they can interact with that are harmless and adequate. Chains with plastic hooks are allowed and a single chain is enough from the standpoint of those countries. The United Kingdom mandates pigs must have permanent access to sufficient amounts of enrichment, but does not specify the types or materials. As in other countries, the use of a single chain is not allowed as environmental enrichment [38].

The European Union has no regulation on luminosity in rearing environments, but it is recommended to maintain the animals at 40 lux for at least 8 h a day. Germany mandates light intensity of 80 lux and natural daylight, while the requirement is only mandatory in Austria when external corrals are deprived of light [16].

In Brazil, normative instruction n. 113, published in 2020, represents the first set of standards related to animal welfare that must be followed by the swine production chain in Brazil. This instruction establishes as mandatory access to an enriched environment, through materials for handling (such as straw, hay, ropes, chains, wood, shavings, rubber, plastic), in addition to the supply of material for nesting for sows 24 h before the expected date of farrowing.

Although the effects of animal welfare on productivity are not the only reason for proper attention and care towards the matter, it does play a major role in the adoption of those practices. Therefore, possible enhanced productivity figures as a driver for producers to be willing to improve the conditions of confinement systems by making them more adequate to the physical and psychological needs of pigs. Poor welfare may lead to negative impacts on production, reproduction, and growth, in addition to increasing the incidence of diseases and resulting in lower quality meat [39].

## 7. Conclusions

The increased attention to animal welfare has fostered decision-making towards improving the physical and psychological quality of livestock, leading to the creation of specific legislation on the subject in many countries. The pig production chain is increasingly involved with new regulation imposed by guidelines on animal welfare worldwide driven by ethical reasons demanded by society and the requirements of the domestic and, above all, the importer market. Brazil has made visible advances in that regard and, although there is still a long path ahead, major progress can already be observed.

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
