# Peer review of "Global and Brazilian Scenario of Guidelines and Legislation on Welfare in Pig Farming"

_animals, 2022, doi:10.3390/ani12192615_

Round 1

Reviewer 1 Report

Dear Authors,

This is a good write-up. However, major issues concerning plagiarism and referencing will need to be addressed to improve the quality of the manuscript and avoid scrutiny from the scientific community. 

1.    Throughout the review there are statements and facts without references. The definition of a review is to investigate published data, and not so much personal opinion thus these need to be carefully reported.  The following lines are examples where references should be added at the end of important statements:

Line 39

Line 43

Line 106-134 point 2: the first four paragraphs have little to no references and yet many various statements are provided. Do they all come from one source?

Line 149-158 No reference

Line 57, 59: Reference error- please use in-text reference style e.g. The reference number can be inserted at the end. It does not read well to use a number when referring to an author’s work. Please check the manuscript throughout for these errors.

Line 105-128

Line 350

2. For a review it's important to investigate plagiarism as so much content is collected from the published output. Although the plagiarism similarity for this manuscript was not too bad (18% on the reviewer's program), the researcher/s needs to deal with some issues of word-for-word plagiarism of long sentences and even paragraphs. This need to be followed by the correct references for those statements. Below are examples that are highlighted:

Line 92

Line 149-159

Line 201-204

Line 228-230

Line 246-247

Line 283-285

Line 289-295

Line 302-309

Line 328-330

 Minor corrections

Line 219-226: Bold, please rectify

Line 375: Please clarify the Brazil, normative instruction no. 113

Reviewer 2 Report

(12): "more productivity for the creation" needs to be reworded.

(22): Remove duplicated word:  "other countries to abolish practices questionable practices".

(57): According to [3], ethics issues regarding 57 the current production methods must be revised in countries: word missing in (3)

(208): "Switzerland prohibits the use of farrowing crates, but allows 208 their temporary use during delivery in some exceptional cases". This sentence is very out of place; you should discuss Switzerland ban of both gestation and farrowing crates earlier in the discussion.

(218): Weaning: Remove bold font for all of this paragraph.

(228): "Piglets are used subjected..."; change the wording as it isn't grammatically correct.

(244): "However,  when performed without anesthesia and analgesia, (word must be placed) induces the acute and chronic pain 245 and changes in behavior, besides promoting the acute activation ..."

(249): A veterinary must be changed to a veterinarian

(273) what does "whole males" mean? Must be reworded.

(322): This paragraph lacks correct punctuation and missing words: "In Brazil, the tail cut must be avoided, however, it can 322 be tolerated when measures to adjust the management and quality of the environment 323 provided in the Normative Instruction are adopted, being only allowed to cut the final 324 third of the tail. tail, up to a maximum of the third day of life."

(358): Replace word ambience by environmental conditions

Overall, I suggest really focusing on the English aspect of the narrative to improve flow and coherency as well as flesh out the paragraph on environmental conditions and enrichment as I didn't see enough discussion of the space and stocking density as well as types of flooring and outdoor access allowed. The physical environment is a key aspect of evaluating swine welfare hence I think your paper will be strenghtened by considering this aspect and how it differs between countries and labelling bodies.

Round 2

Reviewer 2 Report

Much improved, thank you!